# Genetic Aspects of Small for Gestational Age Infants Using Targeted-Exome Sequencing and Whole-Exome Sequencing: A Single Center Study

**DOI:** 10.3390/jcm11133710

**Published:** 2022-06-27

**Authors:** Su-Jung Park, Narae Lee, Seong-Hee Jeong, Mun-Hui Jeong, Shin-Yun Byun, Kyung-Hee Park

**Affiliations:** 1Department of Pediatrics, Pusan National University Hospital, Pusan National University School of Medicine, Busan 49241, Korea; psj0430@gmail.com; 2Department of Pediatrics, Pusan National University Children’s Hospital, Pusan National University School of Medicine, Yangsan 50612, Korea; whitecloud11@hanmail.net (N.L.); ibory830@naver.com (S.-H.J.); maldives8@hanmail.net (M.-H.J.); byun410@gmail.com (S.-Y.B.)

**Keywords:** small for gestational age, targeted exome sequencing, whole exome sequencing

## Abstract

Background: The etiology of small for gestational age (SGA) is multifactorial and includes maternal/uterine-placental factors, fetal epigenetics, and genetic abnormalities. We evaluated the genetic causes and diagnostic effectiveness of targeted-panel sequencing (TES) or whole-exome sequencing (WES) in SGA infants without a known cause. Methods: A prospective study was conducted on newborn infants born with a birth weight of less than the 10th percentile for gestational age between January 2019 and December 2020 at the Pusan National University Hospital. We excluded infants with known causes of SGA, including maternal causes or major congenital anomalies or infections. SGA infants without a known etiology underwent genetic evaluation, including karyotyping, chromosomal microarray (CMA), and TES/WES. Results: During the study period, 82 SGA infants were born at our hospital. Among them, 61 patients were excluded. A total of 21 patients underwent karyotyping and chromosomal CMA, and aberrations were detected in two patients, including one chromosomal anomaly and one copy number variation. Nineteen patients with normal karyotype and CMA findings underwent TES or WES, which identified three pathogenic or likely pathogenic single-gene mutations, namely *LHX3*, *TLK2*, and *MED13L*. Conclusions: In SGA infants without known risk factors, the prevalence of genetic causes was 22% (5/21). The diagnostic yield of TES or WES in SGA infants with normal karyotype and CMA was 15.7% (3/19). TES or WES was quite helpful in identifying the etiology in SGA infants without a known cause.

## 1. Introduction

Small for gestational age (SGA) has been defined either as being below the 10th centile for weight for gestational age or as having a birth weight standard deviation score less than −2 [1,2]. The etiology of SGA is multifactorial and includes maternal/uterine-placental factors, fetal epigenetics, and genetic abnormalities [3,4]. Maternal and uterine placental factors include socioeconomic status, maternal nutrition, smoking, alcohol consumption, and diseases such as preeclampsia, placental infarction, and infection [5]. Twenty to fifty percent of the variations in birth weight can be explained by genetic and epigenetic causes, including chromosomal abnormalities, sequence variants, and epigenetic disturbances [6,7]. The association of SGA fetuses with chromosomal abnormalities is well established [2]. Borrel et al. reported fetuses with isolated SGA below the third centile and with a normal karyotyping, pathological genomic imbalances were up to 10% in additional defects [8]. Previous studies on epigenetic influences, especially DNA methylation disturbance, have also been performed [9]. Numerous genes have been associated with the regulation of human height and implicated in growth disorders [10,11]. Bruna et al. examined 55 patients born SGA with persistent short stature and without and identified the cause of short stature and reported that *IHH*, *NPR2*, *SHOX*, and *ACAN* were associated with growth plate development [12].

To the best of our knowledge, there are no studies on the genetic analysis of Korean SGA infants in the neonatal period. The aim of this study was to identify candidate genetic abnormality that might cause prenatal growth restriction and determine whether a genetic test is a helpful diagnostic approach for SGA infants. Therefore, we used karyotyping, chromosomal microarray analysis (CMA) and targeted-panel sequencing (TES)/whole-exome sequencing (WES), especially in the neonatal period of SGA infants with unknown etiology.

## 2. Patients and Methods

### Patients

A prospective study was conducted on newborn infants born with a birth weight of less than the 10th percentile for gestational age between January 2019 and December 2020 at the Pusan National University Hospital.

Using data from the maternal medical chart, we excluded infants with maternal or placental etiology, including multiple pregnancy, placental insufficiency, preeclampsia/HELLP syndrome, renal insufficiency, autoimmune diseases, and smoking history. We also excluded infants with congenital infection and major congenital anomalies, including ventriculomegaly, omphalocele, and major complex heart anomalies. However, infants with hypotonia /hypertonia and facial dysmorphism with no history of prenatal genetic testing and unknown etiology were included.

Karyotyping and chromosomal microarray analysis (CMA) using cord blood were performed in all patients. When no abnormalities were detected by karyotyping and CMA, the infants born in 2019 and 2020 underwent targeted and WES in the neonatal period as per our policy.

## 3. Targeted Panel and Whole-Exome Sequencing

### 3.1. Targeted Exome Sequencing

Genomic DNA was extracted from peripheral blood leukocytes using the chemagic DNA Blood 200 Kit (PerkinElmer, Waltham, MA, USA). A custom Target Enrichment Kit (Celemics, Inc., Seoul, Korea) was designed to target 30 genes. The selected genes were those associated with short stature (growth-related genes derived from the Online Mendelian Inheritance in Man (OMIM) and MedGen databases) and genes involved in the GH-IGF1 pathway regulation in silico. The following genes were included: *CDC6*, *CDT1*, *CUL7K FGFR1*, *GH1*, *GHR*, *GHRHR*, *GLI2*, *GLI3*, *HESX1*, *IGF1*, *IGF1R*, *KDM6A*, *KMT2D*, *LHX3*, *NIPBL*, *OBSL1*, *ORC1*, *ORC4*, *ORC6*, *PCNT*, *POU1F1*, *PROP1*, *PTPN11*, *RPS6KA3*, *SMARCAL1*, *SMC1A*, *SMC3*, *SOS1*, and *SOX3*. Sequencing was performed on the MiSeq platform (Illumina, Inc., San Diego, CA, USA). Obtained sequence reads were aligned to the hg19 human reference sequencing using the Burrows–Wheeler Aligner software (BWA version 0.7.12, San Diego, CA, USA). For in silico analysis of missense variants, the Sorting Intolerant from Tolerant (SIFT), PolyPhen-2, and Mutation Taster algorithms were used to predict variants that alter protein function. Mean coverage of reading depth was 450 depth and 99.8% of bases on-target. Sequence variants were classified into five categories: pathogenic, likely pathogenic, variants of uncertain clinical significance (VUS), likely benign, and benign, according to the American College of Medical Genetics and Genomics Standards and Guidelines.

### 3.2. Whole-Exome Sequencing

This assay was performed using the PerkinElmer Sciclone^®^ (Waltham, MA, USA) G3 Workstation combined with the Agilent SureSelect Clinical Research Exome capture kit (#G9496A 5190–7344), followed by sequencing of the coding regions and splice sites on the Illumina NextSeq 550 (High-Output v2 kit). Exomes were sequenced to achieve a completeness > 95% of bases covered with at least 15 reads across the entire exome. Reads were aligned to the GRCh37 reference sequence using the Burrows–Wheeler Aligner (BWA 0.7.17, version 0.7.17, San Diego, CA, USA), and variant calls were made using the Genomic Analysis Tool Kit (GATK v4.0.3.0). Variants are subsequently filtered to identify: (1) variants classified as disease-causing mutations in public databases with minor allele frequencies <5.0% in the Genome Aggregation Database (gnomAD, http://gnomad.broadinstitute.org/ accessed on 27 May 2022); (2) nonsense, frameshift, and canonical splice-site variants in disease-associated genes with a minor allele frequency ≤1.0% in gnomAD; and (3) variants with minor allele frequency ≤5.0% in the gnomAD in a patient-specific phenotype-driven gene list. The evidence for phenotype-causality was then evaluated for each variant resulting from the filtering strategies mentioned above, and variants were classified based on the ACMG/AMP criteria (Richards et al., 2015) with ClinGen rule specifications (https://www.clinicalgenome.org/working-groups/sequence-variant-interpretation/, accessed on 27 May 2022). Variants were reported according to HGVS nomenclature (https://varnomen.hgvs.org/, accessed on 27 May 2022). Only variants with evidence for causing or contributing to disease or variants of uncertain significance in genes highly related to the reported patient phenotype were included in the final report. All variants included in this report were confirmed via Sanger sequencing or other orthogonal sequencing techniques.

## 4. Standard Protocol Approval, Registration, and Patient Consent

Ethical approval for this study was granted by the Institutional Review Board of Pusan National University Hospital, and fully informed written consent was obtained from each participant’s parents (2101-002-098).

## 5. Results

During the study period, 82 singleton SGA infants were born at our hospital. Among them, 61 patients were excluded on account of the known causative factors, including maternal preeclampsia (*n* = 21), placental insufficiency (*n* = 15), diabetes mellitus (*n* = 7), autoimmune diseases (*n* = 6), maternal smoking (*n* = 3), congenital structural anomaly (*n* = 4), congenital infection (*n* = 3), and refusal to test (*n* = 2). Finally, 21 patients underwent karyotyping and CMA. Among them, 19 infants with normal karyotype and CMA results were subjected to TES or WES (Figure 1).

The clinical characteristics of 21 patients are shown in Table 1. The mean GA was 35.4 (30.5–41.2) weeks, and the mean BW was 1719 (790–2700 g). The mean percentile of a BW for GA was 3.3. In twelve patients, the BW was below the third percentile of BW for GA. Symmetric and asymmetric type SGAs were observed in 12 and 9 patients, respectively. 

Karyotyping and CMA showed aberrations in 2/21 patients, including one patient with chromosomal anomaly and one with copy number variations (CNVs). SGA15 suffered from respiratory distress syndrome (RDS) and patent ductus arteriosus, which was ligated surgically. Although karyotyping revealed X chromosome deletion and she was diagnosed with Turner syndrome, she did not have typical morphologic features of Turner syndrome in the neonatal period. SGA2 was a preterm infant who suffered from RDS after birth. Karyotyping and CMA revealed heterozygous deletion at 4p, related to Wolf–Hirschhorn syndrome (WHS). Over time, she revealed sucking difficulty and distinctive facial features and was diagnosed with WHS. 

We identified 8 gene mutations in 19 patients with normal karyotype and CMA. Among them, three mutations (SGA 6, 16, and 18) were pathogenic or likely pathogenic (Table 2). SGA 6 was born as a preterm infant and suffered from RDS and necrotizing enterocolitis. On the routine thyroid function test, he was diagnosed with congenital hypothyroidism. TES showed monoallelic *LHX3* mutation (c.935G.A) associated with combined pituitary hormone deficiency (CPHD).

SGA 16 was born as a preterm infant and suffered from RDS, bronchopulmonary dysplasia, and retinopathy of prematurity. WES showed *TLK2* gene mutation, which was associated with mental retardation. She was followed up after discharge from NICU and revealed developmental delay and microcephaly at ten months of age. SGA 18 was born as full term and showed an incomplete imperforated anus. A novel de novo missense mutation, c.5698C > T (p.Arg1900Ter), was identified in *MED13L* associated with MED13L syndrome. He developed hypotonia and developmental delay progressively. He was finally diagnosed with MED13L syndrome.

In SGA 5, 8, 12, 13, and 20, *CDT1* (*n* = 1), *PCNT* (*n* = 3), *KMT2D* (*n* = 1), *FLNB* (*n* = 1), and *OBLS1* (*n* = 1) were identified, respectively, but these genes have been classified to be variants of uncertain significance; none of the parents have been tested for variants identified in our study.

## 6. Discussion

SGA can be the result of both constitutional or fetal growth restriction (FGR), which is defined as a fetus being unable to reach its growth potential [13]. Among genetic causes of SGA, chromosomal anomalies, including trisomy 18 and Turner syndrome, have accounted for up to 19% of fetuses with FGR [8]. However, the incidence of submicroscopic duplications/deletions and single gene disorder in FGR with normal karyotype is not well established.

Array-based genomic copy number analysis has recently become a research tool and a clinical genetic test in the diagnostic work-up in several clinical settings [14]. The detection rates of CMA in FGR patients were 10–18.8% in several studies. Canton et al. analyzed 51 patients with SGA with unknown cause using CMA and found that 8 of 51 patients (16%) had pathogenic or probably pathogenic copy number variants (CNVs) [15]. Hui zhu et al. investigated the clinical value of CMA in 107 FGR patients. Karyotyping identified chromosomal aberration in 9.3%, while CMA detected them in 18.8% of the study population [16].

Borrell et al. performed a meta-analysis to estimate the incremental yield of CMA compared to karyotyping in FGR. They revealed a 4% incremental yield of CMA over karyotyping in non-malformed FGR fetuses and a 10% incremental yield in FGR when associated with fetal malformations. The most frequently found pathogenic CNVs in that study were 22q11.2 duplication, Xp22.3 deletion, and 7q11.23 deletion, particularly in isolated FGR [8]. 

We identified one infant with chromosomal aberration (5%) and one with CNVs (5%) among the 21 patients using karyotyping and CMA simultaneously. These detection rates (2/21 patients, 9.5%) were quite a lot lower than in previous studies (18–19%) [8,14,15,16]. In the present study, we focused on SGA infants without a known cause. We excluded major structural anomalies such as omphalocele. One SGA infant with omphalocele was diagnosed with down syndrome in our study period, but the infant was excluded from the present study because of a major anomaly. Therefore, the rate of genetic abnormalities would be higher if we included all the SGA neonates. Nevertheless, karyotyping and CMA helped establish an early diagnosis in this study. In the case of SGA 2, we might not have been able to diagnose WHS so early without performing CMA because the patient had no typical symptoms and signs for WHS in the neonatal period. Of course, he could have been diagnosed when the symptoms became more distinctive later. Further, SGA 15 may have been eventually diagnosed with Turner syndrome after the clinical symptoms developed progressively. However, we could not diagnose her so early in the neonatal period because she had no typical Turner syndromic features. 

The advent of new genomic technologies, including massively parallel sequencing, has provided a genetic diagnosis for many children with short stature of unknown cause, especially among patients with syndromic conditions [12]. Experience with molecular genetic testing is still limited, but WES revealed approximately 23–50% of genetic variations in cases with normal cytogenetic and CMA results. Susanne et al. reported that genetic analyses in SGA newborns using an array comparative genomic hybridization, genome-wide methylation studies, and exome sequencing. That study identified the genetic abnormality that likely contributed to SGA in 4 of 21 patients (19%) [17]. 

We detected three single-gene mutations in 19 SGA infants with normal karyotype and CMA (15.7%). One infant (SGA 6) showed congenital hypothyroidism, hypoglycemia, cholestasis, and cryptorchidism, which were symptoms of combined pituitary hormone deficiency (CHPD) and prematurity [17]. Therefore, we might not have been able to consider him as CHPD during the hospital days because he was also a preterm infant. Brain MRI was performed after WES to confirm pituitary abnormalities, but his MRI was normal. This variant has been described previously in a patient with CHPD with normal brain MRI [18,19]. *LHX3* mutation in SGA 6 might be a strong candidate gene to cause SGA [20]. However, this patient needed further investigation to confirm CHPD and its potential pathogenic nature. 

In the case of SGAs 16 and 18, both were diagnosed by WES. They would otherwise be diagnosed after developmental delay, and mental retardation had developed more distinctly, thus resulting in a delayed diagnosis. Specifically, infant SGA 18 showed FGR and an incomplete imperforated anus without any other symptoms and signs of MED13L syndrome. Smol et al. reported 36 patients with MED13L molecular anomaly [20]. Among the 36 patients, only one had FGR/IUGR, while no one developed an imperforated anus. All patients had motor, speech, and moderate to severe intellectual delay, and the median age for independent walking was 25 months (ranging from 18 to 30 months). 

In our study of SGA infants without known risk factors, the prevalence of pathological chromosomal or subchromosomal abnormalities was 22% (5/21), similar to the previous report. However, caution should be excised in the interpretation because our study included only SGA without known etiology. As a result, the diagnostic yield of TES or WES in SGA infants was 3/19 (15.7%).

However, there are opposite opinions. Ma Y et al. studied 85 SGA infants without a known cause and concluded that molecular genetic analysis is not recommended for isolated SGA pregnancies without other abnormal findings. In their study, pathological subchromosomal anomalies were detected by CMA or WGS in 10% and 2% of SGA subjects with and without malformation, respectively [21].

Our study has several limitations. First, only a small number of patients were included. The study conclusions, especially regarding the diagnostic yield, cannot be generalized because genetic tests could not be performed on all the SGA infants. Secondly, many previous studies have defined IUGR exclusively by birth weight below the 10th percentile for gestational age without further differentiation between infants that had suffered from prenatal growth restriction and those born solely as SGA. In this study, we could not separate SGA and IUGR distinctly. Some infants born exclusively as SGA constitutionally might have been included in this study, even if they did not need to be tested. 

We conclude that some sequence variants identified TES/WES might contribute to prenatal growth failure, and TES/WES were quite helpful in establishing an early diagnosis in SGA infants with normal karyotype and CMA. Early diagnosis in some patients of this study may have important consequences for the care and counseling of patients and their parents. Further studies are required to know the incidence of genetic causes of SGA and whether such genetic evaluation in the neonatal period can become an effective diagnostic approach in SGA neonates.

## Figures and Tables

**Figure 1 jcm-11-03710-f001:**
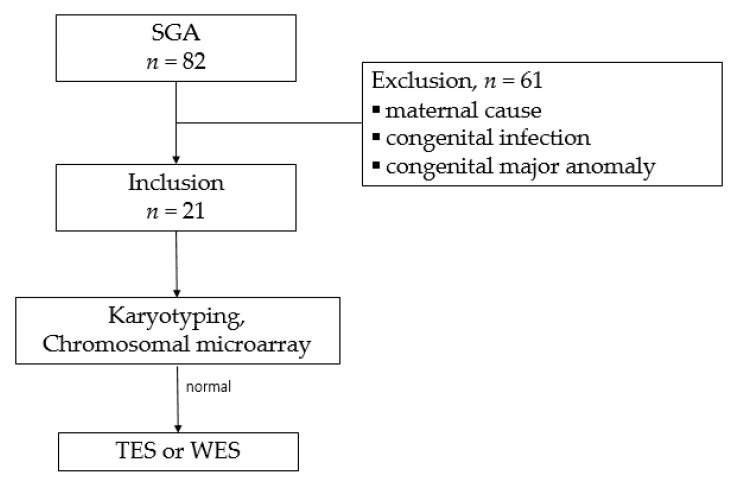
Study selection flowchart.

**Table 1 jcm-11-03710-t001:** Clinical characteristics of patients.

	Sex	Delivery	GA	BW	Percentile of BW	Length	Percentile of Length	HC	Percentile of HC	Ponderal Index	Type of SGA
SGA 1	F	C/S	34 + 4	1093	3	36.5	10	28	10	2.25	Asymmetric
SGA 2	F	C/S	35 + 6	1455	0	42	5	27.5	0	1.96	Symmetric
SGA 3	F	C/S	35 + 5	1795	3	41.5	4	30.5	15	2.5	Asymmetric
SGA 4	F	C/S	39 + 0	2600	5	47	8	32	4	2.5	Symmetric
SGA 5	F	C/S	37 + 1	2280	8	46	25	31.5	14	2.34	Asymmetric
SGA 6	M	C/S	34 + 1	890	0	34.5	0	27.5	1	2.17	Symmetric
SGA 7	M	C/S	33 + 4	1314	1	38	1	28	1	2.36	Symmetric
SGA 8	F	NSVD	37 + 0	1960	2	44	8	31.5	16	2.3	Asymmetric
SGA 9	M	C/S	34 + 5	1764	6	41	3	30.5	21	2.56	Asymmetric
SGA 10	M	C/S	35 + 0	1643	2	41.5	4	30.3	14	2.3	Asymmetric
SGA 11	F	C/S	37 + 0	2110	4	41	0	31.5	16	3.06	Asymmetric
SGA 12	F	C/S	35 + 4	1160	0	35	0	26.5	0	2.7	Symmetric
SGA 13	F	C/S	35 + 5	1950	8	45	33	30	8	2.14	Symmetric
SGA 14	M	C/S	32 + 2	1093	3	36	1	27	4	2.34	Symmetric
SGA 15	F	C/S	32 + 4	1249	0	35.5	0	26.5	0	2.79	Symmetric
SGA 16	F	C/S	30 + 4	790	3	34	2	23.5	0	2.0	Symmetric
SGA 17	M	NVSD	37 + 1	2360	8	44	3	32.5	29	2.77	Asymmetric
SGA 18	M	NSVD	41 + 2	2700	1	48	3	33	3	2.44	Symmetric
SGA 19	F	C/S	34 + 1	1510	5	40	6	27.5	1	2.36	Symmetric
SGA 20	F	C/S	37 + 6	2290	4	44	3	31	4	2.7	Symmetric
SGA 21	F	C/S	37 + 0	2110	4	41	0	31.5	16	3.06	Asymmetric

Abbreviations: SGA, small for gestational age; M, male; F, female; NSVD, normal spontaneous vaginal delivery; C/S, cesarian section; GA, gestational age; BW, birth weight; HC, head circumference.

**Table 2 jcm-11-03710-t002:** Mutations detected in TES/WES.

	Method of Detection	Gene	cDNA Change	Protein Change	Classification	Related Disease or Gene/OMIM Disease
SGA 5	TES	*CDT1*	c.366G > T	p.Glu1224Asp	VUS	Meier-Gorlin syndrome
SGA 6	TES	*LHX3*	c.935G > A	p.Arg312Gln	LPV	Combined pituitary hormone deficiency (CPHD)
SGA 8	TES	*PCNT* *PCNT*	c.3167G > A c.5543A > G	p.Gly1056Asp p.Glu1848Gly	VUS VUS	Microcephalic osteodysplastic primordial dwarfism Microcephalic osteodysplastic primordial dwarfism
SGA 12	TES	*KMT2D* *PCNT*	c.6548A > G c.5647C > T	p.Tyr2183Cys p.Arg1883Trp	VUS VUS	Kabuki syndrome Microcephalic osteodysplastic primordal dwarfism
SGA 13	WES	*FLNB*	c.5959A > C	p.Asn1987His	VUS	Atelosteogenesis, boomerang dsyplasia, Larsen syndrome
SGA 16	WES	*TLK2*	c.31C > T	p.Arg11Ter	LPV	Mental retardation, autosomal dominant 57
SGA 18	WES	*MED13L*	c.5698C > T	p.Arg1900Ter	PV	MED13L syndrome
SGA 20	WES	*OBLS1*	c.2810_2812del	p.Glu937del	VUS	3-M syndrome

Abbreviations: SGA, small for gestational age; cDNA, complementary deoxyribo nucleic acid; OMIM, online mendelian inheritance in man; TES, targeted-exome sequencing; WES, whole-exome sequencing; VUS, variant of uncertain significance; LPV, likely pathogenic variant; PV, Pathogenic variant.

## Data Availability

All data generated or analyzed during this study are included in this published article.

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
