# Peer review of "Genetic Aspects of Small for Gestational Age Infants Using Targeted-Exome Sequencing and Whole-Exome Sequencing: A Single Center Study"

_jcm, 2022, doi:10.3390/jcm11133710_

Round 1

Reviewer 1 Report

Summary: A cohort of babies who are small for gestational age (SGA) from a period of 2 years were screened for genetic abnormalities that may contribute to their undergrowth. Babies with other underlying conditions/factors that are known to lead to SGA were pre-excluded with the final cohort size of 21 babies undergoing firstly cytogenetic testing and if negative molecular testing either by targeted panel and/or whole exome sequencing. The authors found that 5 of these 21 babies had genetic variants that were considered to the primary etiology for the SGA.

General comments: This is overall a clear, well-presented manuscript of the study. Whilst the overall proportion of babies with genetic findings is lower compared to other studies, the specificity of the inclusion criteria and the small size of cohort are likely to be the main contributors to this difference.

1) I was confused on targeted panel vs whole exome sequencing. Did all 19 babies undergo both or just one of these tests? Is there a reason for one or the other (or in other words, is there a particular presentation that meant a baby would be tested using one of these methods over the other)? I think some clarification on this point in the Method and also in Figure 1 might be helpful for the reader.

2) Relating to the targeted panel sequencing, the authors should indicate some performance indicators e.g. mean/median read depth or percentage of gene (coding regions) covered with a minimum read depth.

3) In the methodology of WES, the authors mentioned that a patient-specific phenotype-driven gene list was used as part of the filtering process. The authors should clarify how this list is compiled, e.g. by manual compilation, in silico or AI-assisted for the reader to understand the comprehensiveness of the analysis.

4) Table 2: please provide transcript information for the point variants, as this is a minimum requirement for reporting variants. Please consider including the predicted protein changes to provide the reader some clue of how pathogenic these variants are likely to be.

5) Also on table 2, it may be useful to include your classification of the variants (e.g. pathogenic, uncertain significance) as an additional column for clarification. As your classifications were based on ACMG guidelines, the criteria you've used should be presented as well.

6) The second column in table 2 doesn't have a heading - presumed to be method of detection: but what does DES refer to? And since there are two methods of NGS performed, please specify for these cases whether this is TES or WES as used in the methodology section.

7) Have any of the parents been tested for the variants identified? This was not mentioned in the method/results - even if no segregation testing has been performed, it may be helpful for the reader to know this information, perhaps by adding a line to this effect in the methodology or results.

8) Again on table 2: the columns of 'Related Disease or gene' and 'Clinical diagnosis' overlap significantly and I therefore wonder about the need for the latter. If the authors decide to adopt the above suggestion of listing the pathogenicity of the genetic finding, then the latter is superfluous. Using only the OMIM number (e.g. for TLK2) and a non-standard acronym IIDADFFWOWCD do not give the readers any information and should be replaced by the full OMIM association instead.

9) In your discussion, you mentioned that 'WES revealed approximately 23-50% of chromosomal abnormality in cases with normal cytogenetic and CMA results'. That appears to be a very high proportion of chromosomal abnormalities. Does 'chromosomal abnormality' refer to other copy number and structural variants below the detection limit in other cytogenetic testing, or do you in fact refer to molecular variants (e.g. single nucleotide variants, small insertions/deletions)? If the latter, it may be better to lump these together as 'genetic variations' as a whole and specify the types of variants that form the 23-50%?

Minor changes:

10) Abstract, page 1, line 24: "pathologic and likely pathologic" should be replaced by "pathogenic and likely pathogenic"

11) The gene name LHX3 is misnamed in multiple places: abstract, page 1, line 24; page 4, line 131; page 6, line 194; there may be others - please check thoroughly if there are other instances.

12) Gene names should be italicised across the whole manuscript.

13) Page 4, line 131: the variant name should be c.935G>A rather than "c.935G.A".

14) Table 2: the CDT1 variant name should have a lower case "c." rather than "C.".

15) Table 2: the disease name for KMT2D should be "Kabuki syndrome" rather than "KABUK".

16) Table 2: the OBSL1 variant name should be "c.2810_2812del" rather than "c.2810-2812del"

17) Page 5, line 145-146: where you have stated that "these genes have been confirmed to be variants of uncertain significance" - do you perhaps mean that "the variants identified in the genes have been classified as those of uncertain significance" or similar? The genes themselves are not of uncertain significance.

18) Table 1 - there are abbreviations that have not been spelled out elsewhere. Perhaps these can be spelled out as a footnote at the bottom of the table?

Author Response

We thank the editor and reviewers of the “Journal of Clinical Medicine” for taking their time to review our article. We have made some corrections and clarifications in the manuscript after going over the reviewer’s comments. The changes are summarized below :

Response to Reviewer 1’s comment

Point 1 : I was confused on targeted panel vs whole exome sequencing. Did all 19 babies undergo both or just one of these tests? Is there a reason for one or the other (or in other words, is there a particular presentation that meant a baby would be tested using one of these methods over the other)? I think some clarification on this point in the Method and also in Figure 1 might be helpful for the reader.

Response 1 :

The babies underwent just one of tests (targeted or WES). The babies born at 2020 and 2021 underwent targeted and WES respectively. So we add that in the method.

Before) When no abnormalities were detected by karyotyping and CMA, TES and WES were performed in the neonatal period as per our policy.

After) When no abnormalities were detected by karyotyping and CMA, the infants born at 2019 and 2020 underwent targeted and WES respectively in the neonatal period as per our policy.

Point 2 :  

Relating to the targeted panel sequencing, the authors should indicate some performance indicators e.g. mean/median read depth or percentage of gene (coding regions) covered with a minimum read depth.

Response 2 :

We totally agree with you. Thank you for your advice. We add this in the method like following. 

Mean coverage of read depth was 450 depth and 99.8% of bases on-target.

Point 3 :  In the methodology of WES, the authors mentioned that a patient-specific phenotype-driven gene list was used as part of the filtering process. The authors should clarify how this list is compiled, e.g. by manual compilation, in silico or AI-assisted for the reader to understand the comprehensiveness of the analysis.

Response 3 :

The selected genes were those associated with short stature (growth-related genes derived from the Online Mendelian Inheritance in Man (OMIM) and MedGen databases) and genes involved in the GH-IGF1 pathway regulation by in silico.

Point 4 :  Table 2: please provide transcript information for the point variants, as this is a minimum requirement for reporting variants. Please consider including the predicted protein changes to provide the reader some clue of how pathogenic these variants are likely to be.

Response 4 :

We add the predicted protein changes in table 2 as your commendation.

Point 5 :  Also on table 2, it may be useful to include your classification of the variants (e.g. pathogenic, uncertain significance) as an additional column for clarification. As your classifications were based on ACMG guidelines, the criteria you've used should be presented as well.

Response 5 :

We add the predicted protein changes in table 2 as your commendation.

Point 6 : The second column in table 2 doesn't have a heading - presumed to be method of detection: but what does DES refer to? And since there are two methods of NGS performed, please specify for these cases whether this is TES or WES as used in the methodology section.

Response 6 : We made a mistake in writing. We changed from DES to WES.

Point 7 :  Have any of the parents been tested for the variants identified? This was not mentioned in the method/results - even if no segregation testing has been performed, it may be helpful for the reader to know this information, perhaps by adding a line to this effect in the methodology or results.

Response 7 : Unfortunately none of the parents have been tested. We add the mention about this in the results like following.

And none of the parents have been tested for variants identified in our study.

Point 8 :  ) Again on table 2: the columns of 'Related Disease or gene' and 'Clinical diagnosis' overlap significantly and I therefore wonder about the need for the latter. If the authors decide to adopt the above suggestion of listing the pathogenicity of the genetic finding, then the latter is superfluous. Using only the OMIM number (e.g. for TLK2) and a non-standard acronym IIDADFFWOWCD do not give the readers any information and should be replaced by the full OMIM association instead.

Response 8 : We changed as your advice.

Point 9 :  In your discussion, you mentioned that 'WES revealed approximately 23-50% of chromosomal abnormality in cases with normal cytogenetic and CMA results'. That appears to be a very high proportion of chromosomal abnormalities. Does 'chromosomal abnormality' refer to other copy number and structural variants below the detection limit in other cytogenetic testing, or do you in fact refer to molecular variants (e.g. single nucleotide variants, small insertions/deletions)? If the latter, it may be better to lump these together as 'genetic variations' as a whole and specify the types of variants that form the 23-50%?

Response 9 : We mean latter. So we changed as your advice.

WES revealed approximately 23-50 % of genetic variations in cases with normal cytogenetic and CMA results

Point 10 :  Abstract, page 1, line 24: "pathologic and likely pathologic" should be replaced by "pathogenic and likely pathogenic"

Response 10 : We changed as you pointed out.

Point 11 :  The gene name LHX3 is misnamed in multiple places: abstract, page 1, line 24; page 4, line 131; page 6, line 194; there may be others - please check thoroughly if there are other instances.

Response 11 : We changed as you pointed out.

Point 12 :  Gene names should be italicised across the whole manuscript

Response 12 : We changed as you pointed out.

Point 13 :  Page 4, line 131: the variant name should be c.935G>A rather than "c.935G.A".

Response 13 : We changed as you pointed out.

Point 14 :  Table 2: the CDT1 variant name should have a lower case "c." rather than "C.".

Response 14 : We changed as you pointed out.

Point 15 :  Table 2: the disease name for KMT2D should be "Kabuki syndrome" rather than "KABUK".

Response 15 : We changed as you pointed out.

Point 16 :  Table 2: the OBSL1 variant name should be "c.2810_2812del" rather than "c.2810-2812del"

Response 16 : We changed as you pointed out.

Point 17 :  Page 5, line 145-146: where you have stated that "these genes have been confirmed to be variants of uncertain significance" - do you perhaps mean that "the variants identified in the genes have been classified as those of uncertain significance" or similar? The genes themselves are not of uncertain significance.

Response 17 :

We mean that “the variants identified in the genes have been classified as those of uncertain clinical significance”.

but these genes have been confirmed to be variants of uncertain significance

  • but these genes have been classified to be variants of uncertain significance.

Point 18 :  Table 1 - there are abbreviations that have not been spelled out elsewhere. Perhaps these can be spelled out as a footnote at the bottom of the table?

Response 18 : We changed as you pointed out.

We thank the editor and reviewers of the “Journal of Clinical Medicine” for taking their time to review our article. We have made some corrections and clarifications in the manuscript after going over the reviewer’s comments. The changes are summarized below :

Response to Reviewer 1’s comment

Point 1 : I was confused on targeted panel vs whole exome sequencing. Did all 19 babies undergo both or just one of these tests? Is there a reason for one or the other (or in other words, is there a particular presentation that meant a baby would be tested using one of these methods over the other)? I think some clarification on this point in the Method and also in Figure 1 might be helpful for the reader.

Response 1 :

The babies underwent just one of tests (targeted or WES). The babies born at 2020 and 2021 underwent targeted and WES respectively. So we add that in the method.

Before) When no abnormalities were detected by karyotyping and CMA, TES and WES were performed in the neonatal period as per our policy.

After) When no abnormalities were detected by karyotyping and CMA, the infants born at 2019 and 2020 underwent targeted and WES respectively in the neonatal period as per our policy.

Point 2 :  

Relating to the targeted panel sequencing, the authors should indicate some performance indicators e.g. mean/median read depth or percentage of gene (coding regions) covered with a minimum read depth.

Response 2 :

We totally agree with you. Thank you for your advice. We add this in the method like following. 

Mean coverage of read depth was 450 depth and 99.8% of bases on-target.

Point 3 :  In the methodology of WES, the authors mentioned that a patient-specific phenotype-driven gene list was used as part of the filtering process. The authors should clarify how this list is compiled, e.g. by manual compilation, in silico or AI-assisted for the reader to understand the comprehensiveness of the analysis.

Response 3 :

The selected genes were those associated with short stature (growth-related genes derived from the Online Mendelian Inheritance in Man (OMIM) and MedGen databases) and genes involved in the GH-IGF1 pathway regulation by in silico.

Point 4 :  Table 2: please provide transcript information for the point variants, as this is a minimum requirement for reporting variants. Please consider including the predicted protein changes to provide the reader some clue of how pathogenic these variants are likely to be.

Response 4 :

We add the predicted protein changes in table 2 as your commendation.

Point 5 :  Also on table 2, it may be useful to include your classification of the variants (e.g. pathogenic, uncertain significance) as an additional column for clarification. As your classifications were based on ACMG guidelines, the criteria you've used should be presented as well.

Response 5 :

We add the predicted protein changes in table 2 as your commendation.

Point 6 : The second column in table 2 doesn't have a heading - presumed to be method of detection: but what does DES refer to? And since there are two methods of NGS performed, please specify for these cases whether this is TES or WES as used in the methodology section.

Response 6 : We made a mistake in writing. We changed from DES to WES.

Point 7 :  Have any of the parents been tested for the variants identified? This was not mentioned in the method/results - even if no segregation testing has been performed, it may be helpful for the reader to know this information, perhaps by adding a line to this effect in the methodology or results.

Response 7 : Unfortunately none of the parents have been tested. We add the mention about this in the results like following.

And none of the parents have been tested for variants identified in our study.

Point 8 :  ) Again on table 2: the columns of 'Related Disease or gene' and 'Clinical diagnosis' overlap significantly and I therefore wonder about the need for the latter. If the authors decide to adopt the above suggestion of listing the pathogenicity of the genetic finding, then the latter is superfluous. Using only the OMIM number (e.g. for TLK2) and a non-standard acronym IIDADFFWOWCD do not give the readers any information and should be replaced by the full OMIM association instead.

Response 8 : We changed as your advice.

Point 9 :  In your discussion, you mentioned that 'WES revealed approximately 23-50% of chromosomal abnormality in cases with normal cytogenetic and CMA results'. That appears to be a very high proportion of chromosomal abnormalities. Does 'chromosomal abnormality' refer to other copy number and structural variants below the detection limit in other cytogenetic testing, or do you in fact refer to molecular variants (e.g. single nucleotide variants, small insertions/deletions)? If the latter, it may be better to lump these together as 'genetic variations' as a whole and specify the types of variants that form the 23-50%?

Response 9 : We mean latter. So we changed as your advice.

WES revealed approximately 23-50 % of genetic variations in cases with normal cytogenetic and CMA results

Point 10 :  Abstract, page 1, line 24: "pathologic and likely pathologic" should be replaced by "pathogenic and likely pathogenic"

Response 10 : We changed as you pointed out.

Point 11 :  The gene name LHX3 is misnamed in multiple places: abstract, page 1, line 24; page 4, line 131; page 6, line 194; there may be others - please check thoroughly if there are other instances.

Response 11 : We changed as you pointed out.

Point 12 :  Gene names should be italicised across the whole manuscript

Response 12 : We changed as you pointed out.

Point 13 :  Page 4, line 131: the variant name should be c.935G>A rather than "c.935G.A".

Response 13 : We changed as you pointed out.

Point 14 :  Table 2: the CDT1 variant name should have a lower case "c." rather than "C.".

Response 14 : We changed as you pointed out.

Point 15 :  Table 2: the disease name for KMT2D should be "Kabuki syndrome" rather than "KABUK".

Response 15 : We changed as you pointed out.

Point 16 :  Table 2: the OBSL1 variant name should be "c.2810_2812del" rather than "c.2810-2812del"

Response 16 : We changed as you pointed out.

Point 17 :  Page 5, line 145-146: where you have stated that "these genes have been confirmed to be variants of uncertain significance" - do you perhaps mean that "the variants identified in the genes have been classified as those of uncertain significance" or similar? The genes themselves are not of uncertain significance.

Response 17 :

We mean that “the variants identified in the genes have been classified as those of uncertain clinical significance”.

but these genes have been confirmed to be variants of uncertain significance

  • but these genes have been classified to be variants of uncertain significance.

Point 18 :  Table 1 - there are abbreviations that have not been spelled out elsewhere. Perhaps these can be spelled out as a footnote at the bottom of the table?

Response 18 : We changed as you pointed out.

Reviewer 2 Report

Please see the attached review.

Author Response

We thank the editor and reviewers of the “Journal of Clinical Medicine” for taking their time to review our article. We have made some corrections and clarifications in the manuscript after going over the reviewer’s comments. The changes are summarized below :

Response to Reviewer 2’s comment

Point 1. 1. Introduction- too short, lack of genetic reasons for SGA, lack of the main aim of the study (only description of what was performed).

Response 1.

The aim of this study was to identify candidate genetic abnormality that might cause prenatal growth restriction and determine whether genetic test is a helpful diagnostic approach for SGA infants. Therefore, we used karyotyping, chromosomal microarray analysis (CMA) and targeted panel sequencing (TES) /whole-exome sequencing (WES), especially in the neonatal period of SGA infants with unknown etiology.

Point 2. In the newborns it is rather length not height.

Response 2. We changed from height to length.

Point 3. Lines 130-133 and 191-193: the child with suspected CPHD- the clinical picture is not full. What were the results performed at hypoglycemia (cortisol, GH), IGF1, was brain MRI performed? Usually children with CPHD do not born SGA. It is rather “growth hormone stimulation test”, again usually it is not needed in newborns with CPHD.

Response 3 :

Actually CPHD is difficult to diagnose in neonatal period as you know. The baby was also preterm infant. Brain MRI was performed at 2 months of age. We described the clinical features in discussion like following.

One infant (SGA 6) showed congenital hypothyroidism, hypoglycemia, cholestasis, and cryptorchidism, which were symptoms of combined pituitary hormone deficiency (CHPD) and prematurity.[15] Therefore, we might not have been able to consider him as CHPD during the hospital days because he was also a preterm infant. Brain MRI was performed to confirm pituitary abnormalities, but his MRI was normal. This variant has been described previously in a patient with CHPD with normal brain MRI.[16] LHX3 mutation in SGA 6 might be a strong candidate gene to cause SGA. However, this patient needed further investigation to confirm CHPD and its potential pathogenic nature.

Point 4. Patient SGA 15- unclear (Trisomy 21?). Lines 169-170: “One SGA infant with omphalocele was diagnosed with Down syndrome in our study period.”- is it the same patient? SGA 15 Karyotyping and CMA Trisomy 21, X chromosome deletion Down syndrome, Turner syndrome Turner syndrome

Response 4. The SGA patient with omphalocele is different from SGA 15 patient. The SGA with omphalocele was not included in our study because of major abnormality (omphalocele).

Point 5. Lines 206-208: “In our study of SGA infants without known risk factors, the prevalence of pathological chromosomal or subchromosomal abnormalities was 22% (5/21), similar to the previous report. However, caution should be exercised in the interpretation because previous studies had different cohorts.”- unclear, which previous report?

Response 5. We mean previous studies were reference 11-14 articles. So we changed that like below.

These detection rates (2/21 patients, 9.5%) were quite lower than in previous studies. [11-14]

Point 6. Line 27: “TES or WES was quite helpful in identifying etiology in SGA infants without a known cause.”; line 210: “TES or WES were quite helpful in establishing an early diagnosis in SGA infants with normal karyotype and CMA.”; lines 169-170: “These detection rates were quite lower than in previous studies.”- please modify, “quite” is not informative.

Response 6. We changed as your advice.

Point 7. No clear conclusions given.

Response 7 : We added and changed as your advice.

We conclude that some sequence variants identified TES/WES may contribute to prenatal growth failure and TES/WES were quite helpful in establishing an early diagnosis in SGA infants with normal karyotype and CMA. Early diagnosis in some patients of this study may have important consequences for care and couseling of patients and their parents.

Point 8. References list- too short

Response 8 : we add more references.

Point 9. 238-242, 248-250- unnecessary

Response 9. We don’t know exactly what you mean 238-242, 248-250.

Please let me know, I will delete.

Round 2

Reviewer 2 Report

Please see the attached review.

Author Response

Response to Reviewer 2’s comment

  1. Introduction- please widen the paragraph and include the sort overview of known genetic reasons of SGA

Response : we add and include what you recommend as following.

The association of SGA fetuses with chromosomal abnormalities is well established [2]. Borrel et al. reported fetuses with isolated SGA below the 3rd centile and with a normal karyotyping, pathological genomic imbalances were up to 10% in additional defets [8]. Previous studies on epigenetic influences, especially DNA methylation disturbance, have also been performed [9]. Numerous genes have been associated with the regulation of human height and implicated in growth disorders [10,11]. Bruna et al. examined 55 patients born SGA with persistent short stature without and identified cause of short stature and reperted IHH, NPR2, SHOX and ACAN were associated with growth plate development [12].

  1. The child with suspected CPHD- “LHX3 mutation in SGA 6 might be a strong candidate gene to cause SGA.”- please add a reference.

LHX3 mutation in SGA 6 might be a strong candidate gene to cause SGA [19].

  1. These detection rates (2/21 patients, 9.5%) were quite lower than in previous studies. [11- 14]- please add the information regarding the detection rates in the cited ref. I would recommend not using the phrase “quite” while giving the numbers, percentage etc.

These detection rates (2/21 patients, 9.5%) were quite lower than in previous studies (18-19%) [11-14].

  1. Please clarify why the above numbers changed in respect to the first manuscript version (Lines 206-208: “In our study of SGA infants without known risk factors, the prevalence of pathological chromosomal or subchromosomal abnormalities was 22% (5/21), similar to the previous report.”

We identified one infant with chromosomal aberration (5%) and one with CNVs (5%) among the 21 patients using karyotyping and CMA simultaneously. These detection rates (2/21 patients, 9.5%) were quite lower than in previous studies (18-19%) [11-14].

  • This means 2 patients detected by karyotyping and CMA only.

In our study of SGA infants without known risk factors, the prevalence of pathological chromosomal or subchromosomal abnormalities was 22% (5/21), similar to the previous report.

  • This means 5 patients detected using all methods including not only karyotyping/CMA but also TES/WES.

  1. Paragraph Ethics approval and consent to participate- please add the information about patients’ parents informed consent

All participating patients signed and informed consent from their parents.
